# Applications of Imaging Technologies in Fuchs Endothelial Corneal Dystrophy: A Narrative Literature Review

**DOI:** 10.3390/bioengineering11030271

**Published:** 2024-03-11

**Authors:** Sang Beom Han, Yu-Chi Liu, Chang Liu, Jodhbir S. Mehta

**Affiliations:** 1Saevit Eye Hospital, Goyang 10447, Republic of Korea; m.sangbeom.han@gmail.com; 2Singapore National Eye Centre, Singapore 168751, Singapore; liuchiy@gmail.com; 3Singapore Eye Research Institute, Singapore 168751, Singapore; chang.liu@seri.com.sg; 4Department of Ophthalmology, Yong Loo Lin School of Medicine, National University of Singapore, Singapore 119228, Singapore

**Keywords:** anterior segment imaging, anterior segment optical coherence tomography, corneal tomography, Fuchs endothelial corneal dystrophy, in vivo confocal microscopy, specular microscopy

## Abstract

Fuchs endothelial corneal dystrophy (FECD) is a complex genetic disorder characterized by the slow and progressive degeneration of corneal endothelial cells. Thus, it may result in corneal endothelial decompensation and irreversible corneal edema. Moreover, FECD is associated with alterations in all corneal layers, such as thickening of the Descemet membrane, stromal scarring, subepithelial fibrosis, and the formation of epithelial bullae. Hence, anterior segment imaging devices that enable precise measurement of functional and anatomical changes in the cornea are essential for the management of FECD. In this review, the authors will introduce studies on the application of various imaging modalities, such as anterior segment optical coherence tomography, Scheimpflug corneal tomography, specular microscopy, in vitro confocal microscopy, and retroillumination photography, in the diagnosis and monitoring of FECD and discuss the results of these studies. The application of novel technologies, including image processing technology and artificial intelligence, that are expected to further enhance the accuracy, precision, and speed of the imaging technologies will also be discussed.

## 1. Introduction

Among the five layers of the human cornea, epithelium, that is, Bowman’s layer, stroma, Descemet membrane, and endothelium, the corneal endothelium is the innermost layer of the cornea and comprises a single layer of corneal endothelial cells, which plays a key role in the maintenance of corneal transparency by the action of Na^+^/K^+^ pumps in the corneal endothelial cells [1]. Extensive loss or damage of corneal endothelial cells may lead to endothelial decompensation, which subsequently results in corneal edema and opacity [1]. Corneal decompensation can be caused by various causes, such as trauma, toxins, drugs, and surgical injury, among which Fuchs endothelial corneal dystrophy (FECD) is one of the most common diseases that affect corneal endothelium [2], with an estimated prevalence of 7.33% [3]. FECD is a slow-onset complex genetic disorder characterized by the progressive degeneration of the corneal endothelium, which eventually results in irreversible corneal edema and vision loss [4]. Corneal transplantation is still the mainstay of treatment for advanced FECD cases [5], among which posterior lamellar keratoplasty, such as Descemet’s stripping automated endothelial keratoplasty (DSAEK) or Descemet’s membrane endothelial keratoplasty (DMEK), has been emerging as the most frequently performed corneal transplant procedure in patients with FECD [6]. Descemet stripping only (DSO) has been suggested to be a viable treatment option for certain cases of FECD [7]. In addition to these surgical methods, novel treatment methods, such as cell injection therapy, gene therapy, and pharmacological-associated treatment, have also been introduced [8,9]. Progression of FECD is associated with changes in corneal endothelium, including the formation and accumulation of guttae, reduced endothelial cell density (ECD), and alteration of normal endothelial cell morphology [10,11]. It can also lead to changes in other corneal layers, such as the accumulation of extracellular matrix in the Descemet membrane (DM), thickening of the DM, stromal scarring, subepithelial fibrosis, and epithelial bullae formation [12,13].

Therefore, imaging devices that can enable visualization and precise assessment of corneal layers, particularly corneal endothelium, are critical for the diagnosis and management of FECD. Advances in technology have led to the development of anterior segment imaging modalities that can be useful for the monitoring of FECD, such as anterior segment optical coherence tomography (AS-OCT), Scheimpflug corneal tomography, specular microscopy, and in vitro confocal microscopy.

In this narrative review, we aim to provide an overview of the application of anterior segment imaging technology in FECD and discuss the potential future development of these modalities with advancements in imaging processing technology and artificial intelligence (AI). For this literature review, a non-systematic search for relevant scientific articles was conducted in December 2023 from the PubMed/MEDLINE database using the following keywords: “Fuchs endothelial corneal dystrophy”, “FECD”, “corneal endothelial dystrophy”, “corneal imaging”, “anterior segment optical coherence tomography”, “corneal tomography”, “Scheimpflug imaging”, “in vivo confocal microscopy”, “IVCM”, “specular microscopy”, “corneal retroillumination”, and “retroillumination photography”, which was limited to literature in English. Among the retrieved studies, those evaluated to be relevant to imaging technologies in FECD were included in this review.

## 2. Anterior Segment Optical Coherence Tomography (AS-OCT)

AS-OCT, based on low-coherence interferometry technology, enables real-time cross-sectional imaging of anterior segment structures [14,15]. It is a valuable tool for the diagnosis and monitoring of corneal diseases, as it can provide detailed images of each corneal layer [15,16,17]. Thus, it can be helpful for the visualization of DM and the corneal endothelial layer and the diagnosis and management of FECD (Figure 1) [15].

AS-OCT can be a valuable tool for the assessment of attachment of endothelial graft after DSAEK or DMEK [16,17,18,19,20,21], which may be important for decision-making regarding re-attachment procedures in the early postoperative period [16,17,18,19,20,21].

Augustin et al. [22] recently showed that both AS-OCT and Scheimpflug imaging (Pentacam^®^ HR; OCULUS Optikgeräte GmbH, Wetzlar, Germany) were able to detect the tomographic signs of subclinical corneal edema, such as displacement of the thinnest point of the cornea, loss of regular isopachs, and the presence of posterior surface depression, with good agreement between the two devices in 72.3% of the patients with FECD. Using the Heidelberg Spectralis spectral domain (SD) AS-OCT, Huang et al. [23] showed that the presence of corneal guttae was significantly associated with increased mean Descemet’s membrane thickness (DMT) and central corneal thickness (CCT). They also revealed that both DMT and CCT had a significant correlation with density in guttae evaluated using a specular microscope (NSP-9900; Konan Medical Inc., Hyogo, Japan) [23]. Yasukura et al. [24] proposed a new severity grading system for FECD using AS-OCT based on pachymetry and posterior elevation maps, which showed 100% inter-observer agreement. Wertheimer et al. [25] also recently introduced a method of quantitative evaluation of the severity of FECD based on corneal stromal optical densities in the AS-OCT images. In this study, the increased corneal optical densities on AS-OCT were significantly correlated with increased corneal edema and decreased visual acuity in patients with FECD [25].

Swept-source (SS) AS-OCT is also shown to be useful for the detection of characteristic features of subclinical FECD, such as increased thickness of the thinnest corneal point, higher order Fourier indices, 3 and 6 mm asymmetry, and a posterior ectasia screening index [26]. SS AS-OCT also revealed that advanced FECD was associated with decreased posterior keratometry flatness, steepness, average eccentricity, 3 and 6 mm posterior spherical indices, and increased corneal apex thickness [26]. Arnalich-Montiel et al. [27]. also showed that both the SS AS-OCT and the Scheimpflug camera could measure corneal thickness in patients with FECD with high repeatability and reproducibility, whereas the SS AS-OCT was superior to the Scheimpflug camera in the repeatability and reproducibility of the measurements in all corneal locations in both untreated eyes and eyes with an endothelial graft [27].

In 2010, Shousha et al. [28] demonstrated that ultra-high resolution (UHR) AS-OCT can visualize DM and measure its thickness. In this study, the DM in young and elderly participants without any known eye conditions appeared as a single smooth opaque line and a band of two smooth opaque lines using their custom-built UHR AS-OCT [28]. In eyes with FECD, the thickening of the band of two opaque lines was observed, with a change in the posterior line to a wavy and irregular appearance [28]. The UHR AS-OCT also showed that DM in FECD was substantially thicker than that in young and elderly participants without any known eye conditions (34 ± 11 μm vs. 10 ± 3 μm and 16 ± 2 μm, respectively; *p* < 0.001) [28]. UHR-OCT enables visualization of endothelial grafts after endothelial keratoplasty [29], which is important for the postoperative management of FECD.

Iovino et al. [30] attempted to evaluate corneal endothelium features in FECD using a preliminary three-dimensional (3D) AS-OCT visualizing the inner surface of the corneal center. They determined the OCT pattern based on the level of corneal endothelial reflectivity and the signal distribution and showed that the reflectivity level had a positive correlation with the clinical severity grade of FECD scored using a slit lamp biomicroscope and a negative correlation with the integrity of corneal endothelium, respectively [30]. Eleiwa et al. [31] generated 3D maps of endothelium–Descemet’s membrane complex thickness (En-DMT) of the central 6 mm cornea using high-definition OCT imaging (Envisu R2210; Bioptigen, Buffalo Grove, IL, USA). This study demonstrated that the mean En-DMT of central, paracentral, and peripheral corneal regions had a high correlation with the clinical severity of FECD, suggesting that the analysis of the 3D AS-OCT images can be a reliable tool for monitoring the severity of FECD [31].

The application of AI in the interpretation of AS-OCT images may improve the detection and management of FECD [32,33]. In 2020, Eleiwa et al. [32] introduced A deep learning (DL) algorithm for automated diagnosis and staging of FECD based on high-definition OCT imaging (Envisu R2210; Bioptigen, Buffalo Grove, IL, USA), which showed excellent performance for detecting early and late-stage FECD and discriminating healthy corneas from all FECD (area under the curve [AUC] 0.997, 0.974, and 0.998; sensitivity 91%, 100%, and 99%; and specificity 97%, 92%, and 98%, respectively). Bitton et al. [33] also developed a DL model for the detection of corneal edema using AS-OCT images. This model showed a high AUC (0.97) for the detection of significant edema (defined as 20 μm), which can be helpful for decision-making for endothelial keratoplasty [33]. Treder et al. [34] introduced a DL model using a convolutional neural network (CNN) for automated quantification of graft dislocation after DMEK in AS-OCT images, which showed excellent reliability with an accuracy of 96%, a sensitivity of 98%, and a specificity of 94%.

However, AS-OCT is not yet capable of imaging the corneal endothelium at a cellular level or determining the morphology and density of the corneal endothelial cells [35]. Moreover, it cannot provide detailed visualization of each corneal layer at a cellular or histologic level. Thus, technological advancement, including enhancing resolution to the cellular level and precise imaging of the corneal endothelial surface, is needed for the clinical application of the device. 

## 3. Scheimpflug Corneal Tomography

Scheimpflug corneal tomography has emerged as a valuable tool for the diagnosis and monitoring of FECD [36]. It can be helpful for the detection of wavefront errors and changes in corneal curvature associated with FECD (Figure 2) [37,38,39]. In 2015, Wacker et al. [37] demonstrated that both anterior and posterior high-order aberrations (HOAs) and backscatter calculated using Scheimpflug images increase even in the early stage of FECD. They also showed that the advancement of FECD might be associated with abnormal posterior toricity because of a more marked increase in horizontal than vertical corneal thickness using Scheimpflug images [38]. In this study, posterior corneal power was less negative in moderate and advanced FECD, which might explain the hyperopic shift after DMEK [38].

In 2017, Chu et al. [39] showed that the mean area density and ratio of DM density versus area density evaluated using Scheimpflug imaging were elevated, suggesting that the Pentacam system might provide an objective, quantitative method of detecting and monitoring the progression of FECD.

In 2020, Patel et al. [40] reported that subjective interpretations of Scheimpflug tomography images in FECD showed high inter- and intra-observer agreement for disease classification, and clinically significant disagreement was uncommon. In 2019, Sun et al. [41] revealed that Scheimpflug tomography can detect subclinical corneal edema that cannot be detected using slit-lamp biomicroscopy, which can be helpful for determining the candidate for endothelial keratoplasty. In 2020, Patel et al. [42] showed that the Scheimpflug tomography pachymetry map and posterior elevation map could be useful for the prediction of the prognosis of FECD. They revealed that the risk of FECD progression increases according to the number of parameters present, which include loss of regular isopachs, displacement of the thinnest point of the cornea, and the presence of posterior surface depression.

In 2018, Mingo-Botín et al. [43] demonstrated that Pentacam corneal thickness maps showed good reproducibility and repeatability in patients with FECD both before and after endothelial keratoplasty, indicating the feasibility of the Pentacam system for monitoring FECD.

In 2021, Zander et al. [44] developed a prediction model for the extent of corneal edema resolution based on the five variables evaluated using Scheimpflug tomographic imaging, i.e., focal posterior depression, nonparallel isopachs, anterior and posterior corneal backscatter, and central corneal thickness, which might be helpful for identification of patients who could benefit from endothelial keratoplasty.

However, Scheimpflug corneal tomography has the limitation that it is not capable of precise visualization of changes in cellular or tissue level in each corneal layer, although it can enable functional evaluation of corneal endothelial health and assessment of the progression and treatment response of FECD [36].

## 4. Specular Microscopy

Specular microscopy is a non-invasive technique that allows for in vivo visualization of corneal endothelium by using specular reflection with slit-lamp biomicroscopy [15,45]. It enables morphological assessment of the corneal endothelium because it can determine endothelial cell density (ECD), polymegathism represented by coefficient of variation (CV), pleomorphism represented by hexagonality, CCT, and the presence and progression of guttae (Figure 3) [11,15,45].

Even in early cases without corneal edema, FECD can be associated with impairment of the quality of vision (QOV) due to light scatter and glare caused by corneal guttae [46]. In 2015, Watanabe et al. [47] showed that the area ratio of the corneal guttae (ARCG) in the corneal endothelial cells measured using multifocal specular microscopy had a correlation with the deterioration of QOV in patients with mild FECD without corneal edema on slit-lamp examination, suggesting that specular microscopy can be a useful tool for monitoring the progression of mild FECD. 

In 2014, Fujimoto et al. [48] measured the percentages of abnormal corneal endothelial areas in specular microscopy images of corneas with FECD. They reported that the corneal endothelial damage was more severe in the central zone than in the peripheral zone, and the percentage of the abnormal corneal endothelial region only in the peripheral zone had a significant correlation with the grade of FECD [48].

A slit-scanning wide-field contact specular microscope can provide a panoramic view of the corneal endothelial layer from the surgical limbus to the limbus (Figure 4) [49]. In 2019, Lee et al. [50] showed that the analysis of ECD using slit-scanning wide-field contact specular microscopy (CellChek C, version 1.0.0.5; Konan Medical, Nishinomiya, Japan) showed no difference from that using non-contact specular microscopy, suggesting that the device can be a useful tool for the monitoring of FECD. 

Smartphone-based specular microscopy is expected to be applied for screening abnormalities of corneal endothelium in rural and underdeveloped countries [51]. In 2021, Mantena et al. [52] developed an image processing method for smartphone-based specular microscopy using a smartphone attached to a digital slit lamp that allowed for automated computation of ECD, CV, and hexagonality values, which achieved good concordance with the values evaluated with conventional specular microscopy.

The application of artificial intelligence in the analysis of specular microscopy images is expected to enable precise monitoring of corneal endothelial diseases. Researchers have developed various AI-based methods for automated segmentation and quantitative assessment of corneal endothelial cells in specular microscopic images [53,54,55,56,57,58]. DL Methods based on CNN U-net have been shown to improve the speed and accuracy of morphologic analysis of specular microscopic images compared to manual evaluation or conventional software [53,54,57].

In 2019, Daniel et al. [59] introduced a method based on U-net for automated segmentation and quantitative analysis of corneal endothelial cells using specular microscopy images, which showed good agreement with the manual annotation. Vigueras-Guillén et al. [60] showed that a U-net convolutional neural network-based method improved the precision and accuracy of estimation of ECD, CV, and hexagonality compared to conventional specular microscopy. Moreover, they reported that the method enabled the segmentation and analysis of an image in a few minutes [60]. They also reported that their AI-based automated method has better performance with improved accuracy and prevision than conventional software in the estimation of ECD, CV, and hexagonality from specular microscopy images of the cornea after ultrathin DSAEK [61].

Okumura et al. [62] developed a method using CNN U-Net for the segmentation of corneal endothelial cell borders and guttae in specular microscopy images of a mouse FECD model (Col8a2L450W/L450W knock-in mice). This method showed strong agreement with the manual determination for the analysis of the area and number of guttae and ECD, CV, and hexagonality, suggesting its potential as a tool for objective and fast analysis of corneal endothelial abnormalities in FECD. In 2021, Shilpashree et al. [63] also introduced a method for automatic segmentation of corneal endothelial cells in specular microscopy images using the combination of the modified U-Net and Watershed algorithms, which allowed for precise segmentation and analysis of corneal endothelial cells both in healthy and FECD eyes. They revealed that the average perimeter length of the corneal endothelial cells, which represents the length of the paracellular fluid flux pathway into the stroma, increased with the percentage of guttae in FECD [63], suggesting that the AI-based analysis of specular microscopy images can be useful for monitoring the subtle progression of FECD. In 2022, Vigueras-Guillén et al. [64] proposed a deep learning method for improving the accuracy of segmentation of corneal endothelial cells occluded by guttae in specular microscopy images of FECD. They reported that their protocol using DenseUNets with feedback non-local attention showed significantly decreased error compared to conventional software in the estimation of the corneal parameters, including ECD, CV, and hexagonality in specular microscopy images with guttae [64].

The major limitation of specular microscopy is that the acquisition of reliable images is impossible in advanced FECD cases with serious corneal edema and endothelial cell loss due to increased light scattering [11], although it is a valuable tool for evaluation of the corneal endothelial layer in FECD with no or mild corneal edema [15]. As it can only provide images of the corneal endothelial layer, complementary use of other imaging devices may be necessary for the evaluation of changes in other corneal layers associated with FECD. 

## 5. In Vivo Confocal Microscopy (IVCM)

IVCM is a non-invasive image acquisition technique that enables real-time analysis of all layers of the cornea at the histologic level. Refs. [65,66,67,68,69] are capable of providing a clear image of corneal endothelial cells and guttae [67]. Theoretically, IVCM can be particularly valuable in FECD because it is able to visualize corneal endothelium through corneal edema or haze [65,66,67,68], as its confocal optical arrangement precludes defocusing of light [65]. As Kaufman et al. [70] first described in 1993, guttae appear as dark round bodies sized 20–400 μm, occasionally accompanied by central white reflexes sized 5–10 μm in IVCM [66,68,70].

In 1998, Mustonen et al. [66] described various IVCM findings of corneas with FECD in a case series study including 17 patients. They demonstrated that FECD was associated with various pathological changes in all corneal layers, although the posterior layers are more frequently affected [66]. Pathologic changes observed using IVCM included epithelial bullae, abnormalities in Bowman’s layer, such as the absence of nerves and diffuse light reflection, lesions in the stroma, such as lacunae and diffuse bright reflection due to edema, thickening of the DM and dark bands in the DM, endothelial guttae, and decreased ECD (Figure 5) [66]. Using IVCM, Amin et al. [71] also revealed that abnormalities in the anterior cornea, i.e., increased anterior corneal backscatter, decreased stromal cell density, and an increased number of abnormal subepithelial cells, begin in early-stage FECD, even before the onset of clinically significant edema.

Both IVCM and specular microscopy can be used for the measurement of ECD, and several studies have shown that there was no significant difference in ECD evaluated using either technique in normal cornea or early-stage FECD with minimal corneal edema [66,67,72,73,74,75,76].

However, studies have suggested that IVCM was superior to specular microscopy for the visualization and evaluation of the corneal endothelial layer in advanced FECD with significant corneal edema [67,76,77]. A study including 11 eyes with FECD showed that clear images of corneal endothelium in which determination of ECD was possible were obtained in all 11 eyes (100%) using IVCM, although clear images were obtained only in 4 eyes (36.4%) using specular microscopy, indicating the superiority of IVCM for the assessment of corneal endothelial cells in FECD. Similarly, in another study that included 49 eyes with FECD, ECD grading was precluded due to poor image quality in specular microscopy images of 20 (40.8%) eyes but only in 4 (8.2%) confocal images. A study including 115 eyes with FECD also reported that clear specular microscopy images were obtained only in 27 eyes (23.5%), whereas all ICVM images were clearly obtained in all 115 eyes (100%) [76]. Moreover, in 33 eyes with late-stage FECD, specular microscopy images were precluded in 29 eyes (87.9%) [76]. In this study, Ong Tone et al. [76] also demonstrated that there is a significant decrease in mean ECD in areas surrounding guttae (1296 ± 560 cells/mm^2^ [2]) compared to non-guttae areas (1926 ± 674 cells/mm^2^ [2]) in eyes with FECD using IVCM, suggesting the possible association between guttae and apoptosis of corneal endothelial cells [15]. In 2017, Syed et al. [78] showed peripheral ECD measured using IVCM was the best predictor of severity of the advanced FECD compared with other clinical markers, such as central ECD, CCT, visual acuity, and clinical disease grade.

IVCM can also be useful for monitoring corneal change, not only at the surgical interface but also at the anterior cornea, after endothelial keratoplasty. Baratz et al. [79] used the IVCM for quantitative determination of corneal subepithelial reflectivity (backscatter) and revealed that visual function after DSEK is more affected by residual anterior subepithelial haze than the interface haze. They also showed that younger age was correlated with greater improvement in subepithelial reflectivity after endothelial keratoplasty, which had a correlation with improvement in forward light scatter [79]. Patel et al. [80] reported that IVCM showed decreased cell density of the anterior stroma and the presence of abnormal subepithelial cells in corneas with FECD before surgery, which did not recover after recovery of corneal endothelial function after DSEK, suggesting that abnormality of the anterior cornea associated with FECD might affect the postoperative visual outcome. In 2021, Schiano-Lomoriello et al. [81] compared the IVCM characteristics of FECD after DSAEK with those after DMEK at 6 months after surgery. They found that only DSAEK patients showed a third reflectivity peak in the Z-scan curves, which could be associated with a worse visual outcome than DMEK [81].

IVCM has also been applied for the evaluation of alterations in sub-basal corneal nerves and dendritiform cell density associated with FECD [72,82,83,84]. IVCM of the central corneas demonstrated that the total number of subbasal corneal nerves and main nerve trunks was significantly decreased in early-stage FECD and had a correlation with ECD [72]. Subsequently, Bucher et al. [83] reported that the increasing severity of FECD was significantly correlated with decreased total corneal nerve length and number and a reduced number of both main nerve trunks and branches. In a study by Aggarwal et al. [82], IVCM demonstrated a decreased number and total length of subbasal corneal nerves in both early- and late-stage FECD, which corresponds to reduced corneal sensation in FECD. This study also showed increased dendriform cell density in early FECD, suggesting that an increased immune reaction might have an association with the pathophysiology of FECD [82]. In 2021, Dikmetas et al. [84] also showed that corneal subbasal nerve plexus density measured using IVCM decreased in correlation with the advancing stage of FECD, which might lead to reduced corneal sensitivity in these patients. In 2022, Gillings et al. [85] reported a negative correlation between the severity of FECD and both the length and density of corneal nerve fibers in FECD patients with transcription factor 4 gene (*TCF4*) trinucleotide repeat expansion, whereas this correlation was not found in FECD patients without *TCF4* trinucleotide repeat expansion. These findings suggest that the TCF4 trinucleotide repeat expansion disorder might be classified as a neurodegenerative disease [85].

Ahuja et al. [86] revealed that a reduction in subbasal nerves with abnormal branching observed in IVCM in corneas with FECD could persist even 36 months after endothelial keratoplasty, which might account for decreased corneal sensitivity after recovery of endothelial function in these patients. In 2021, Abicca et al. [87] compared IVCM findings of corneas with FECD after DSAEK with those after DMEK. They reported that there was no significant difference in the number, length, density, and tortuosity of subbasal nerve fibers between the two types of endothelial keratoplasty, although the DSAEK groups showed a lower corneal beading density than the DMEK group [87].

The application of image processing technology and AI is expected to enhance the reliability and speed of the analysis of IVCM images [88,89]. In 2018, Al-Fahdawi et al. [88] introduced a fully automated system for segmentation and quantitative analysis of endothelial cells in IVCM images. Their system, termed the corneal endothelium analysis system (CEAS), enabled automated analysis, including enhancing image quality, segmentation of endothelial cell boundaries, and quantification of the morphological parameters of the corneal endothelial cells in only 6 s per image [88]. The CEAS system showed good agreement with manual analysis [88]. In 2022, Qu et al. [89] developed a method based on deep learning for the automated assessment of corneal endothelial cells in IVCM images. ECD determined using this AI-based method showed good agreement with manually calculated ECD, indicating the potential feasibility of AI for the analysis of IVCM images [89].

However, IVCM has a limitation in the large-scale assessment of the corneal endothelial layer, such as panorama view, due to its small field of view (<0.25 mm^2^ [2]) [90]. To overcome this limitation, methods of reconstruction of wide-field IVCM composite images using a compilation of multiple IVCM images have been developed [91,92,93]. Allgeier et al. [94] introduced an automated method for complete imaging of the sub-basal nerve plexus by combining axial focus plane oscillations and rapid expansion of the acquired area using guided eye movements, which is expected to enable three-dimensional visualization of a wide area of the cornea at the same time. Further development of imaging techniques for IVCM that enable large-scale assessment of each corneal layer would be necessary for a better understanding of FECD.

## 6. Analysis of Retroillumination Photography

Retroillumination photography is an easy and simple method of obtaining images of the corneal endothelial layer with cameras attached to a slit lamp (Figure 6). Several studies introduced methods for the analysis of retroillumination photography for monitoring and treatment of FECD [95,96,97,98].

In 2006, Gottsch et al. [95] developed a method for determining the number and location of guttae and areas with confluent guttae in retroillumination photography of FECD patients using NIH ImageJ software. They analyzed retroillumination photographs taken 23 to 30 months apart in 4 patients with FECD, which confirmed that individual guttae persisted in their original position once they formed and that new guttae can appear within only 2 years [95]. They also suggested that the determination of areas with confluent guttae using their method might be useful for quantitative evaluation of the stage of the FECD [95].

In 2015, Eghrari et al. [96] attempted a manual analysis of the numbers and distribution of guttae in retroillumination photography images. In this study, they showed that the increasing numbers of guttae had a significant correlation with the advancing stage of FECD [91]. Subsequently, they developed an automated analysis method using ImageJ software for objective assessment of retroillumination photography images [96]. Each retroillumination photograph was pre-processed for optimal isolation of guttae and automated calculation of the number of guttae, of which the results showed a strong correlation with manual counts and the Krachmer scale [97].

In 2021, Soh et al. [98] introduced an automated image analysis system for the segmentation and quantitative evaluation of guttae in retroillumination photography images of patients with FECD, which showed good reliability and reproducibility for measurement of the number and size of guttae. Moreover, large-scale assessment using this system could selectively detect the corneal endothelium region significantly affected by confluent guttae, which might be helpful for the determination of the descemetorhexis zone [98].

Retroillumination photography can enable visualization of each gutta by light scattering using only slit lamp microscopy [95]. As it does not require expensive devices, it is an economical and simple method of evaluating corneal guttae. The major limitation of retroillumination photography is that it cannot provide detailed visualization of corneal endothelial cells due to its low image resolution and quality [95].

## 7. Conclusions

In this review, we provided an overview of the application of various imaging technologies for the visualization and assessment of changes in the cornea in patients with FECD. Advances in anterior segment imaging technology have enabled precise and accurate evaluation of morphological and functional alterations in the cornea, even at a cellular level. Imaging techniques, including AS-OCT, Scheimpflug corneal tomography, specular microscopy, IVCM, and retroillumination photography, may improve the diagnostic performance and treatment outcome of FECD by providing detailed information regarding pathologic changes in the cornea caused by the disease. Although imaging technologies were mostly focused on the visualization of corneal endothelium in the previous studies, we also showed that changes in other corneal layers associated with FECD can be assessed using devices including AS-OCT, Scheimpflug corneal tomography, and ICVM. Moreover, we introduced methods of improving the diagnostic efficacy of these devices using novel computer software and AI. A complementary application of these devices might be recommended for the management of FECD because each modality has its own advantages and limitations (Table 1).

The results of the studies introduced in this review suggest that the integration of image processing technology and AI into imaging technology may further enhance the reliability of these technologies, which is critical for the optimal management of FECD.

## Figures and Tables

**Figure 1 bioengineering-11-00271-f001:**
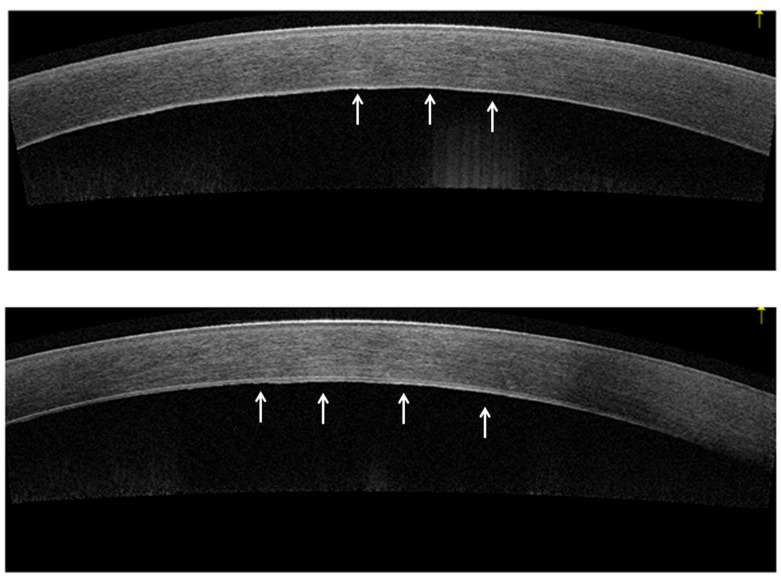
Cross-sectional AS-OCT images of FECD. AS-OCT displays the wavy and irregular appearance of the corneal Descemet Membrane (white arrow).

**Figure 2 bioengineering-11-00271-f002:**
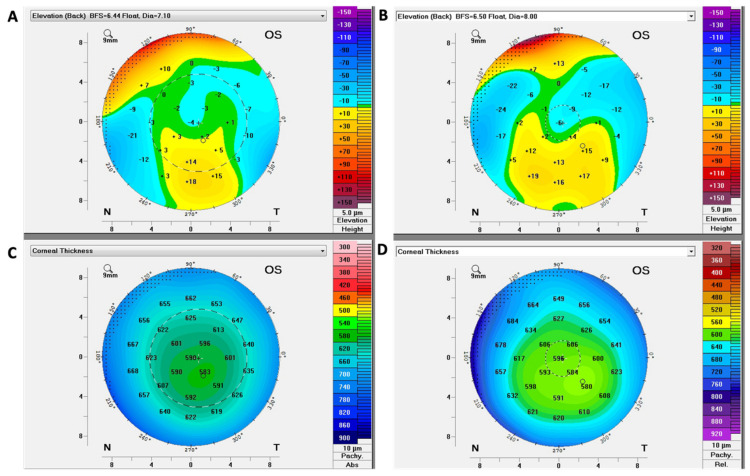
Scheimpflug images of FECD. Posterior elevation maps show focal areas of posterior surface depression at baseline (**A**), and 3 years after the baseline scans show a trend towards negative elevation (**B**). Pachymetry map at baseline (**C**) showing displacement of the thinnest point and loss of isopach concentricity, and 3 years after the baseline scans demonstrating increased corneal thickness (**D**).

**Figure 3 bioengineering-11-00271-f003:**
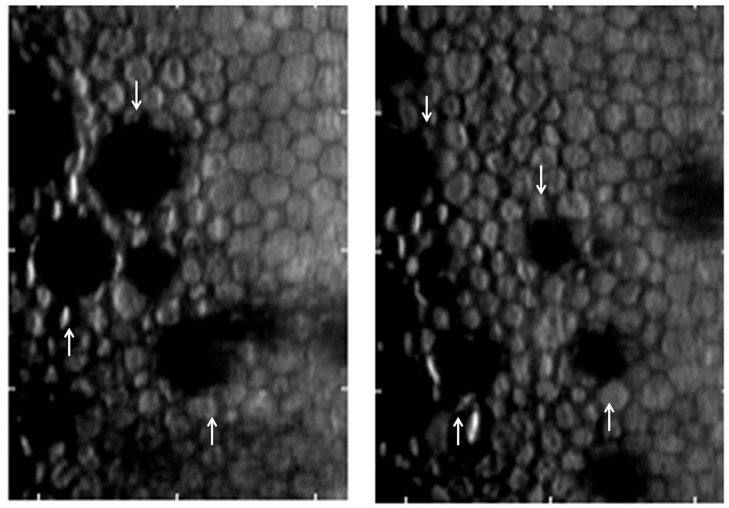
Specular microscopy images of FECD show droplet-shaped guttae (white arrow).

**Figure 4 bioengineering-11-00271-f004:**
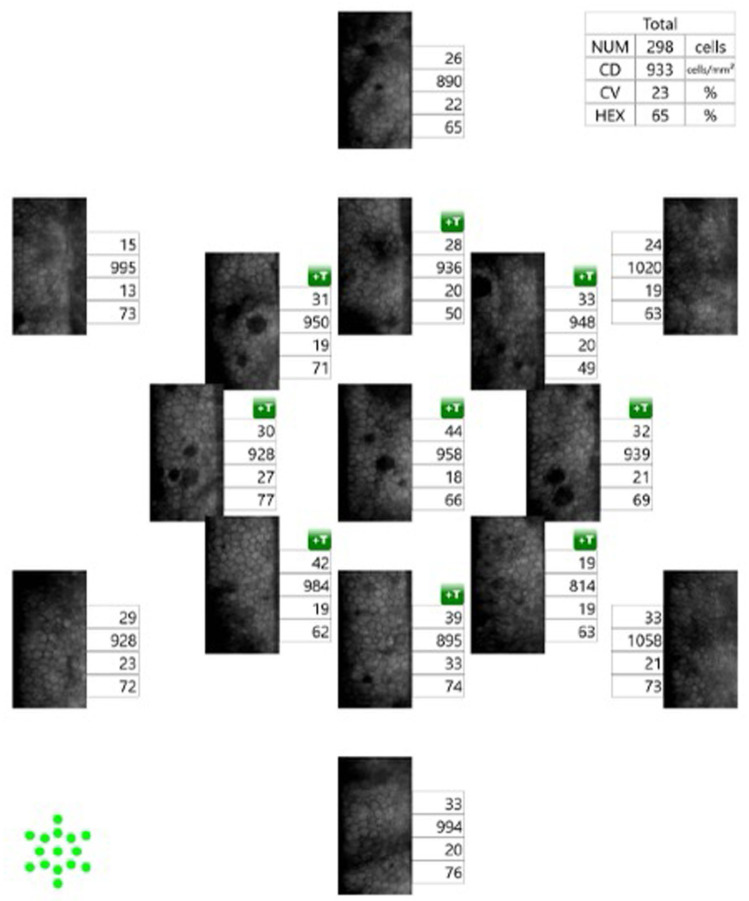
Representative image of non-contact wide-field specular microscopy (CEM-530, Nidek Co., Ltd., Gamagori, Japan) of an FECD patient. Each scan captures one central, eight paracentral (1.3 mm away from the center), and six peripheral (7.3 mm away from the center) images.

**Figure 5 bioengineering-11-00271-f005:**
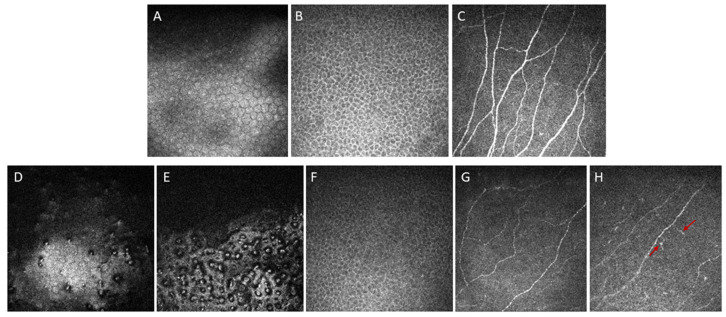
In vivo confocal microscopy images of a healthy subject and FECD. (**A**) Hexagonal endothelium; (**B**) regular shape of basal corneal epithelium; and (**C**) densely anastomosed subbasal nerve plexus in a healthy subject. In a moderate FECD case, the hexagonal shape of the corneal endothelium can still be seen, although the cell margin is not clear (**D**). Both moderate (**D**) and severe (**E**) cases present with corneal guttate with wart-like excrescences in DM. In FECD, the epithelial cells swell with irregular cell morphology (**F**), decreased nerve density (**G**), and the presence of dendritic cells ((**H**), red arrows).

**Figure 6 bioengineering-11-00271-f006:**
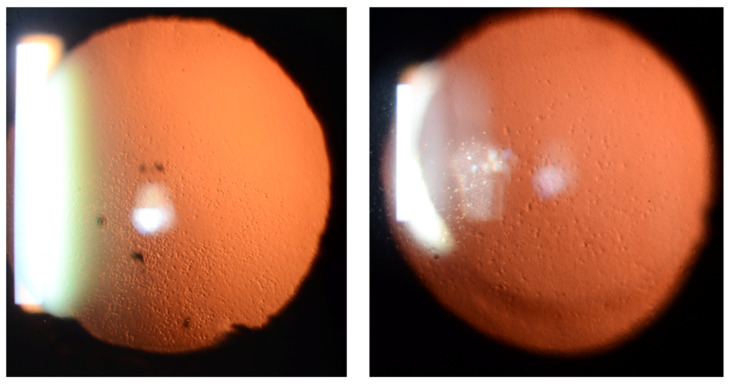
Retroillumination photographs of FECD show diffuse drop-like guttata.

**Table 1 bioengineering-11-00271-t001:** Comparison of various imaging modalities for Fuchs endothelial corneal dystrophy (FECD).

	Anterior Segment Optical Coherence Tomography	Scheimpflug Corneal Tomography	Specular Microscopy	In Vivo Confocal Microscopy	Retroillumination Photography
Tomography of corneal layers	Possible	Possible	Not possible	Not possible	Not possible
Evaluation of changes of corneal curvature associated with FECD	Possible	Possible	Not possible	Not possible	Not possible
Visualization of corneal endothelial cells	Not possible	Not possible	Possible	Possible	Not possible
Measurement of endothelial cell density	Not possible	Not possible	Possible	Possible	Not possible
Visualization of guttae	Not possible	Not possible	Possible	Possible	Possible
Applicable in corneas with serious edema	Possible	Possible	Not possible	Possible	Not possible
Cost	Expensive	Expensive	Expensive	Expensive	Cheap

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
