# Peer review of "Applications of Imaging Technologies in Fuchs Endothelial Corneal Dystrophy: A Narrative Literature Review"

_bioengineering, 2024, doi:10.3390/bioengineering11030271_

Round 1

Reviewer 1 Report

Comments and Suggestions for Authors

This paper aims to review the current imaging techniques to segment the corneal endothelium layers. But it’s just a list of some imaging techniques that have been used for cornea in vitro imaging. And I am not sure why the author chose those imaging techniques, since it doesn’t look like a very comprehensive search for a review paper. The author doesn't seem to know how to write a review paper. 

  1. Overall the review is poorly organized, and doesn’t offer too much value for FECD segment imaging or FECD in vitro diagnosis. The whole structure of this manuscript is just, some year, somebody et al., his research results is xxxx. This is just a reading summary, not a review paper. 

  2. Even though the title says anterior segment imaging, but the whole manuscript doesn’t discuss segmentation at all. Even the last table doesn’t mention which imaging modality can segment the corneal epithelial layer. 

  3. The author should introduce some anatomy of corneal endothelial layers. And if this review aims to focus on FECD, then there should be more discussion about this disease in the introduction, i.e. how does the corneal layer change? The introduction is incomplete.  

  4. Figure 1 is not referenced in the main text. Figure 1 is not labeled properly. What is the difference between those two images? And Figure 1 should have some labels for each layer of cornea. What are those structures? 

  5. All the imaging modalities mentioned in this paper can image the cornea, but which one of them performs cornea segmentation? The title of this paper is “anterior segment imaging in….”. I don’t see any discussion about segmentation. The review is totally irrelevant to segmentation. 

Author Response

This paper aims to review the current imaging techniques to segment the corneal endothelium layers. But it’s just a list of some imaging techniques that have been used for cornea in vitro imaging. And I am not sure why the author chose those imaging techniques, since it doesn’t look like a very comprehensive search for a review paper. The author doesn't seem to know how to write a review paper.  

⇒ We are afraid that there is a misunderstanding regarding this issue, which must be due to our unclear description. As a matter of fact, we did not intend to ‘segment’ the corneal layers. We aimed to provide an overview of application of various imaging techniques in Fuchs endothelial corneal dystrophy (FECD). Regarding the term “anterior segment”, it means structures of anterior segment of the eye, such as, cornea and conjunctiva. Thus, we intended to describe the application of imaging techniques of cornea in patients with FECD, rather than segmentation of the corneal layers. To avoid this misunderstanding, we changed the title to “Application of imaging technologies in Fuchs endothelial corneal dystrophy (FECD): a narrative literature review. 

1) Overall the review is poorly organized, and doesn’t offer too much value for FECD segment imaging or FECD in vitro diagnosis. The whole structure of this manuscript is just, some year, somebody et al., his research results is xxxx. This is just a reading summary, not a review paper.                                                                                                                                                 

 ⇒ We totally agree with the reviewer that this paper may not seem well organized. It is because this paper is neither a systemic review nor meta-analysis. This paper is a literature review that provides an overview of current application of various imaging devices for visualization and assessment of cornea int patients with Fuchs endothelial corneal dystrophy (FECD). To avoid misunderstanding, we added ‘a narrative literature review’ to the Title. 

2) Even though the title says anterior segment imaging, but the whole manuscript doesn’t discuss segmentation at all. Even the last table doesn’t mention which imaging modality can segment the corneal epithelial layer.

 ⇒ We are afraid that there is a misunderstanding regarding this issue, which must be due to our unclear description. As a matter of fact, we did not intend to ‘segment’ the corneal layers. We aimed to provide an overview of application of current imaging techniques to visualize and assess changes in cornea associated with FECD. Regarding the term “anterior segment”, it means structures of anterior segment of the eye, such as, cornea and conjunctiva. Thus, we intended to describe the application of imaging techniques of cornea in patients with FECD, rather than segmentation of the corneal layers. To avoid this misunderstanding, we changed the title to “Application of imaging technologies in Fuchs endothelial corneal dystrophy (FECD): a narrative literature review.

3)     The author should introduce some anatomy of corneal endothelial layers. And if this review aims to focus on FECD, then there should be more discussion about this disease in the introduction, i.e. how does the corneal layer change? The introduction is incomplete.   

⇒ Corneal endothelium is one of the five layers of the cornea, which includes epithelium, Bowman’s layer, stroma, Descemet membrane, endothelium. Anatomically, FECD is mainly associated with pathologies in corneal endothelium. However, it affects all the 5 layers of the cornea. As the reviewer suggested, we added the description regarding the changes in each corneal layer associated with FECD in the introduction (Lines 49-54). Moreover, we also added that the application of various imaging technologies can enable the assessment of changes in each corneal layer, not only the changes in the endothelial layer (Lines 432-435).   

4)   Figure 1 is not referenced in the main text. Figure 1 is not labeled properly. What is the difference between those two images? And Figure 1 should have some labels for each layer of cornea. What are those structures?  

⇒ We are sorry for the mistake. We referenced the Fig. 1 in the revised manuscript (Line 77). Fig. 1 is photos of AS-OCT that shows the changes in Descemet Membrane, one of the 5 layers of the cornea, associated with FECD. As the reviewers requested, we added ‘white arrows’ to indicate the Descemet Membrane. Fig. 1. Shows the wavy and irregular appearance of the Descemet Membrane, as we described in the revised Legend of the Fig. 1.

5) All the imaging modalities mentioned in this paper can image the cornea, but which one of them performs cornea segmentation? The title of this paper is “anterior segment imaging in….”. I don’t see any discussion about segmentation. The review is totally irrelevant to segmentation.  

⇒ Again, we are afraid that there is a misunderstanding regarding this issue, which must be due to our unclear description. As a matter of fact, we did not intend to ‘segment’ the corneal layers. We aimed to provide an overview of application of current imaging techniques to visualize and assess changes in cornea associated with FECD. Regarding the term “anterior segment”, it means structures of anterior segment of the eye, such as, cornea and conjunctiva. Thus, we intended to describe the application of imaging techniques of cornea in patients with FECD, rather than segmentation of the corneal layers. To avoid this misunderstanding, we changed the title to “Application of imaging technologies in Fuchs endothelial corneal dystrophy (FECD): a narrative literature review.

⇒ Thank you very much for your time and efforts. 

Reviewer 2 Report

Comments and Suggestions for Authors

The paper “Anterior Segment Imaging in Fuchs’ Endothelial Corneal Dystrophy” reports on a review concerning the different imaging modalities for diagnosis and monitoring the mentioned disease.

The paper has several issues that should be corrected. The list is presented below:

  1.   The literature review must be updated with recent publications between 2022 and 2024. In the introduction section, the authors should discuss the prevalence of this disease. As there are reported reviews of the same topic, the authors should present the original contributions and discuss how this paper differs from other published review papers ( see, for instance, [1]).

  2. The methodology concerning the search of publications must be reported. In addition, a plot concerning the evolution of the number of publications by year should be presented. Why are the publications cited relevant? What were the criteria used for citing the references?

  3. The authors should briefly discuss the treatment of this disease. The authors should present the eye anatomy and the affected membrane anatomy.

  4. The physical principle concerning the acquisition of each imaging type should be discussed. When an imaging modality includes several sub-modalities, these sub-modalities should also be presented. 

  5. The presentation of different imaging modalities could be more interesting as the authors only describe several publications concerning each modality. However, the presentation of each imaging modality needs to be more tutorial, and the advantages and limitations of each modality should be better discussed. The authors should include a schema showing the hardware for acquisition. A discussion should also be presented concerning the software used for processing the images and visualization.

  6. A discussion section should include the advances, challenges, limitations, and possible future research directions concerning this technology. The conclusion section must be improved.

  7. A comparative table of different imaging modalities concerning the technical aspects and cost should also be included.

  8. Figure 1 is not cited. The authors should mark in the images the details described in the caption and the text (is it Fig 2?). Similar to the rest of the figures

  9. Figure 4 is too small to see the text and image details.

[1] Ong Tone, S., & Jurkunas, U. (2019, May). Imaging the corneal endothelium in Fuchs corneal endothelial dystrophy. In Seminars in ophthalmology (Vol. 34, No. 4, pp. 340-346). Taylor & Francis.

Comments on the Quality of English Language

 Moderate editing of English language required

Author Response

The paper “Anterior Segment Imaging in Fuchs’ Endothelial Corneal Dystrophy” reports on a review concerning the different imaging modalities for diagnosis and monitoring the mentioned disease. 

The paper has several issues that should be corrected. The list is presented below: 

1) The literature review must be updated with recent publications between 2022 and 2024. In the introduction section, the authors should discuss the prevalence of this disease. As there are reported reviews of the same topic, the authors should present the original contributions and discuss how this paper differs from other published review papers ( see, for instance, [1]).

⇒ As the reviewer recommended, we updated the literature review by adding 11 recent publication (Ref#3-13 in the revised manuscript). We also discussed the prevalence of FECD (Line 38) and added relevant reference (#3).  We also added the discussion regarding the original contribution of this paper and how this paper differs from previously papers (Lines 432-437) Briefly, this paper showed that changes in other corneal layers associated with FECD can be assessed using devices including AS-OCT, Scheimpflug corneal tomography and ICVM. Although previous papers mostly focused on the imaging on corneal endothelium.  Moreover, we introduced methods of improving the diagnostic efficacy of these devices using novel computer software and AI, which may further enhance the reliability of these technologies, which is critical for optimal management of FECD.  

2) The methodology concerning the search of publications must be reported. In addition, a plot concerning the evolution of the number of publications by year should be presented. Why are the publications cited relevant? What were the criteria used for citing the references?

⇒ As the reviewer suggested, we described the methodology how we searched for the relevant studies and the criteria for including the articles relevant to this narrative literature review, as follows (Lines 64-71) : For this literature review, a non-systematic search for relevant scientific articles was con-ducted in December 2023 from the PubMed/MEDLINE database using the following key words: “Fuchs endothelial corneal dystrophy”, “FECD”, “corneal endothelial dystrophy”, “corneal imaging”, ”anterior segment optical coherence tomography”, “corneal tomogra-phy”, “Scheimplug imaging”, “in vivo confocal microscopy”, “IVCM”, “specular micros-copy”, “corneal retroillumination”, and “retroillumination photography”, which was lim-ited to literature in English. Among the retrieved studies, those evaluated to be relevant to imaging technologies in FECD were included in this review.

3)    The authors should briefly discuss the treatment of this disease. The authors should present the eye anatomy and the affected membrane anatomy.

⇒ As the reviewer suggested, we added discussion about the treatment of the FECD and relevant references (Lines 41-48, Ref# 3-9). We also added anatomy of the human cornea (Lines 30-31) and affected membrane anatomy associated with FECD (Lines 49-54, Ref#10-13).  

4)     The physical principle concerning the acquisition of each imaging type should be discussed. When an imaging modality includes several sub-modalities, these sub-modalities should also be presented.  

⇒ We also believe the discussion about the physical principle regarding the acquisition of each imaging type in various imaging technologies would be very helpful to the readers. However, as clinical ophthalmologists, it is difficult for us to provide detailed explanation regarding the physical principles of each device and sub-modalities of each technique. Thus. this paper is focused on the clinical application of the imaging devices in patients with FECD, rather than technical principles of the imaging technology. We are sorry to the reviewer.

5)  The presentation of different imaging modalities could be more interesting as the authors only describe several publications concerning each modality. However, the presentation of each imaging modality needs to be more tutorial, and the advantages and limitations of each modality should be better discussed. The authors should include a schema showing the hardware for acquisition. A discussion should also be presented concerning the software used for processing the images and visualization.

⇒ We are sorry to the reviewer because it is difficult for us to provide tutorial, schema and processing of images regarding each imaging technologies as we are clinical ophthalmologist and it is beyond our expertise to discuss technological details. However, as the reviewer suggested, we improved discussion regarding advantages and limitation of each modality (Lines 153-158; 193-196; 270-275; 379-388; 418-423).  

6) A discussion section should include the advances, challenges, limitations, and possible future research directions concerning this technology. The conclusion section must be improved.

⇒ As the reviewer suggested, we improved the conclusion sections including the advances, challenges, limitations (described in Table 1), and future direction of the imaging technologies (Lines 437-439; 441-443) We also described the advantages and limitations of each devices (Lines 153-158; 193-196; 270-275; 379-388; 418-423).  

7) A comparative table of different imaging modalities concerning the technical aspects and cost should also be included.

⇒ Regarding the technical aspect, we are sorry that it is difficult to us to compare the technical differences between each modality. Regarding the cost, we revised the Table 1 to include the comparison of cost between each device. Retroillumination photography is relatively cheap as it does not require additional device other than slit lamp, while other technologies require expensive devices. We also mentioned it in the revised manuscript (Lines 418-422).

 8) Figure 1 is not cited. The authors should mark in the images the details described in the caption and the text (is it Fig 2?). Similar to the rest of the figures

⇒ We are sorry for the mistake. We referenced the Fig. 1 in the revised manuscript (Line 77). Fig. 1 is photos of AS-OCT that shows the changes in Descemet Membrane, one of the 5 layers of the cornea, associated with FECD. As the reviewers requested, we added ‘white arrows’ to indicate the Descemet Membrane. Fig. 1. shows the wavy and irregular appearance of the Descemet Membrane, as we described in the revised Legend of the Fig. 1. We also added the “white arrow” to Fig. 3. to indicate the guttae more clearly.   

9)  Figure 4 is too small to see the text and image details.ll. Even the last table doesn’t mention which imaging modality can segment the corneal epithelial layer.

⇒ As you suggested, we improved the resolution and size of Fig. 4 for better understanding of the text and image detail. Regarding the term “segment”, we are afraid that there is a misunderstanding regarding this issue, which must be due to our unclear description. As a matter of fact, we did not intend to ‘segment’ the corneal layers. We aimed to provide an overview of application of current imaging techniques to visualize and assess changes in cornea associated with FECD. Regarding the term “anterior segment”, it means structures of anterior segment of the eye, such as, cornea and conjunctiva. Thus, we intended to describe the application of imaging techniques of cornea in patients with FECD, rather than segmentation of the corneal layers. To avoid this misunderstanding, we changed the title to “Application of imaging technologies in Fuchs endothelial corneal dystrophy (FECD): a narrative literature review.

⇒Thank you very much for the supportive and pertinent comments.

Round 2

Reviewer 1 Report

Comments and Suggestions for Authors

The previous questions were addressed. 

Reviewer 2 Report

Comments and Suggestions for Authors

The paper has been improved according to the indicated corrections

Comments on the Quality of English Language

Moderate modifications are necessary